# Assessing the efficacy of river-based ocean alkalinity enhancement for carbon sequestration under high emission pathways

Xiao-Yuan Zhu<sup>1</sup>, Shasha Li<sup>1</sup>, Wei-Lei Wang<sup>1\*</sup>

<sup>1</sup> State Key Laboratory of Marine Environmental Science & College of Ocean and Earth Sciences, Xiamen University, Xiamen 361102, China

Abstract. Among various proposed geoengineering methods, ocean alkalinity enhancement (OAE) stands out as a unique solution. By mimicking natural weathering processes, OAE can simultaneously enhance oceanic carbon uptake and mitigate ocean acidification. However, the full efficacy and potential side effects of OAE remain to be fully understood. To evaluate the efficacy of OAE through natural pathways via rivers, we applied a 5-fold alkalinity flux increase (OWE5) at the mouths of global rivers from 2020 to 2100 in a fully coupled Earth System Model under a high-emission scenario (SSP585). In additional sensitivity tests, the flux was increased to 7.5- (OWE75), 10-fold (OWE10), or restored to the control level (OWE0) in 2050. Compared to the control run, global mean surface pH increased by 0.02, 0.03, 0.04, and 0.006; the oceanic inventory of dissolved inorganic carbon (DIC) increased by 5.39, 7.41, 9.50, and 2.06 Pmol; and atmospheric CO<sub>2</sub> concentration decreased by 29, 40, 51, and 11 ppmv under OWE5, OWE75, OWE10, and OWE0, respectively, by the end of the century. The most significant responses to OAE were observed in coastal regions, as well as in the Indian and North Atlantic Oceans. Our simulations demonstrate that OAE via rivers is an effective and practical method, however, even a tenfold increase in alkalinity flux is insufficient to reverse the trends of ocean acidification or rising atmospheric CO2 levels under a high-emission scenario. This underscores the urgent need for complementary technological innovations and aggressive emission reduction strategies to curb CO2 emissions.

Short Summary. Ocean Alkalinity Enhancement (OAE) the only Carbon Dioxide Removal methods that can simultaneously absorb CO2 and alleviate ocean acidification. In this study, we evaluated the effectiveness of riverine OAE under high emission scenario in a fully coupled Earth System Model. The simulations show the riverine OAE effectively boosts ocean carbon uptake and partially combats ocean acidification, but continuous OAE is necessary to achieve the desired outcomes.

## 1 Introduction

Since the Industrial Revolution, rising atmospheric carbon dioxide (CO<sub>2</sub>) levels have driven dramatic climate change—one of the greatest challenges we are facing today. Global average surface temperature has already increased by 1.1 °C relative to the 1850–1900 baseline (IPCC, 2023) and continues to rise, approaching the Paris Agreement's target of limiting warming to below 1.5 °C by the end of this century (UNFCCC, 2015). Excessive warming has triggered

<sup>\*</sup>Correspondence to: Wei-Lei Wang (weilei.wang@xmu.edu.cn)

increasingly frequent and intense extreme events, such as marine heatwaves, polar ice melt, and sea level rise, posing severe risks to both human societies and marine ecosystems. As a major carbon sink, the ocean has absorbed approximately one-third of anthropogenic CO<sub>2</sub> emissions since the Industrial Revolution (Friedlingstein et al., 2023). However, this uptake has led to ocean acidification, which lowers the saturation state of calcium carbonate and threatens calcifying organisms such as corals, foraminifera, and other marine species (Beaufort et al., 2011; Kleypas et al., 1999; Riebesell et al., 2000; Zeebe et al., 2008). Whether through global warming or ocean acidification, the impacts of CO<sub>2</sub> emissions on the climate system are becoming increasingly severe, underscoring the urgent need to reduce atmospheric CO<sub>2</sub> concentrations.

45

Emission reduction and carbon sequestration are two complementary strategies for lowering atmospheric CO<sub>2</sub> concentrations and should be pursued in tandem to effectively mitigate climate change and minimize socioeconomic impacts. For emission reduction, technological innovation is needed to transition away from high-emission energy sources. However, an additional CO<sub>2</sub> sequestration requirement of -5.3 GtCO<sub>2</sub> per year is needed on the base of -2.1 GtCO<sub>2</sub> per year in 2011-2020 even under 76% greenhouse gas emission reduction (Smith et al., 2024). This underscores that deep emission cuts alone will not suffice to achieve carbon neutrality. Therefore, additional technologies must be developed to actively remove CO<sub>2</sub> from the atmosphere—a set of approaches collectively known as Carbon Dioxide Removal (CDR).

To date, several CDR methods have been proposed and evaluated, both theoretically and at small experimental scales, for their effectiveness in removing atmospheric CO<sub>2</sub> (Keller et al., 2014). As the largest carbon reservoir at the Earth surface, the ocean holds substantial potential for enhanced CO<sub>2</sub> uptake. This has led to increasing interest in marine-based CDR (mCDR) approaches. Proposed mCDR strategies include large-scale afforestation of coastal and marine ecosystems (Duarte et al., 2022; Wang et al., 2023), artificial ocean upwelling (Jürchott et al., 2023), ocean alkalinity enhancement (OAE; Eisaman et al., 2023; Oschlies et al., 2023; Renforth & Henderson, 2017), and micronutrient fertilization (Bach et al., 2023; Lampitt et al., 2008). Among these, OAE is promising because it offers the dual benefit of reducing atmospheric CO<sub>2</sub> and direct effect on alleviating ocean acidification, making it an ideal candidate for mitigating CO<sub>2</sub>-driven climate impacts through mCDR.

Alkalinity is defined as the excess of proton acceptors over proton donators in seawater. Approximately 95% of seawater alkalinity is contributed by the carbonate system, primarily in the form of carbonate and bicarbonate ions. A decline in surface alkalinity, driven by enhanced upper-ocean stratification and bio-activity, has been shown to reduce oceanic carbon uptake (Barrett et al., 2025; Chikamoto et al., 2023; Kwiatkowski et al., 2025). OAE works by introducing carbonate, bicarbonate, or other H<sup>+</sup> acceptors into surface waters, thereby increasing carbonate ion concentrations, raising pH, and reducing the partial pressure of CO<sub>2</sub> (*p*CO<sub>2</sub>) in seawater. By altering the air-sea CO<sub>2</sub> disequilibrium, OAE can enhance oceanic CO<sub>2</sub> uptake in

undersaturated regions and reduce outgassing in oversaturated regions, thereby increasing net ocean carbon storage and ultimately lowering atmospheric CO<sub>2</sub> concentrations.

75

Many laboratory and field experiments have assessed the carbon capture potential of OAE, and the biological feedback associated with it (González-Santana et al., 2023; Guo et al., 2023; Montserrat et al., 2017). A wide range of materials has been tested as alkalinity sources, including sodium carbonate, powdered lime, olivine sand, and steel slag. For instance, a mesocosm experiment using sodium carbonate/bicarbonate salts was conducted in coastal waters to investigate potential effects on trace metal cycling and phytoplankton physiology (González-Santana et al., 2023). The results indicated that sodium salt addition did not alter iron dynamics. Another study applying olivine and steel slag to coastal waters found that these materials increased alkalinity by 29 and 361 μmol kg<sup>-1</sup>, respectively, which enhanced CO<sub>2</sub> storage capacity in seawater by 0.9% and 14.8% (Guo et al., 2023). Regarding alkalinity stability, experiments have shown that CO<sub>2</sub>-equilibrated alkaline solutions pose the lowest risk of alkalinity loss to the deep ocean (Hartmann et al., 2023). However, such studies primarily demonstrate local and theoretical outcomes. Comprehensive understanding of the global ocean's response to OAE remains limited.

To address this, recent studies have employed Earth System Models (ESMs) to explore OAE responses at a global scale. For example, a global-scale ocean circulation model coupled with a biogeochemical module simulating olivine dissolution suggested that OAE helps oceanic carbon sequestration, with a particle size of ~1 µm required for full dissolution before sinking to the deep ocean (Köhler et al., 2013). Another centennial-scale simulation tested a fixed 2:1 ratio of alkalinity addition to CO<sub>2</sub> emissions, demonstrating that such alkalinization could maintain surface ocean pH and carbonate chemistry near present-day values (Ilyina et al., 2013). A study using a fully coupled ESM found that applying 0.25 Pmol Alk yr<sup>-1</sup> from 2020 to 2100 could reverse ocean acidification and offset atmospheric CO<sub>2</sub> increases under low-emission (RCP2.6) scenarios, though significantly more alkalinity would be needed under high-emission (RCP8.5) scenarios (Lenton et al., 2018). However, most model-based studies assume a uniformly 100 distributed and constant-rate alkalinity addition across the global ocean—an assumption that is unlikely to be achievable in practice. Although there are also a growing number of regional OAE simulations in recent years (Burt et al., 2021; Feng et al., 2017; He & Tyka, 2023), we still lack research using more practical delivery methods, such as river-based OAE.

On geological timescales, alkalinity is naturally added to the ocean via chemical weathering of rocks on land, with river systems acting as the primary delivery pathway. Though slow, this natural process plays a crucial role in regulating Earth's long-term climate. To mimic this mechanism, we use an emission-driven, fully coupled Earth System Model to evaluate a riverine-based, global-scale OAE scenario under a high-emission pathway (Shared Socioeconomic
 Pathway 5-8.5, SSP585), which reflects a natural and spatially realistic pathway of alkalinity delivery that differs from the commonly assumed uniform ocean-wide input. The responsive CO<sub>2</sub>

concentration configuration in our simulation allows the interactions between climate, ocean chemistry and carbon fluxes and captures the features not captured in prescribed-CO<sub>2</sub> simulations. Specifically, we simulate enhanced natural weathering by increasing riverine alkalinity fluxes by factors of 5, 7.5, and 10. To investigate the potential for climate rebound and the persistence of CO<sub>2</sub> uptake following the cessation of OAE, we halt alkalinity addition in 2050 and extend the simulation through 2100. Finally, we assess how this natural weathering-based OAE strategy affects oceanic CO<sub>2</sub> uptake, surface carbonate chemistry, and the spatial redistribution of alkalinity and dissolved inorganic carbon (DIC). This study provides the transient responses of ocean system to OAE and insights into persistence and reversibility of OAE-induced changes, as well as the suggestion to future study and deployment of OAE.

### 2 Methods

120

135

## 2.1 Model description

This investigation uses Community Earth System Model 2 (CESM2,

https://www.cesm.ucar.edu/models/cesm2), a state-of-the-art, community-developed, fully-coupled earth system model consisting of ocean, atmosphere, land, sea-ice, river, and wave models through a coupler to exchange states and fluxes (Danabasoglu et al., 2020). The atmospheric component in this study is the Community Atmosphere Model Version 6 (CAM6) with a general resolution of 0.9°×1.25°. The ocean component uses the Parallel Ocean Program
Version 2 (POP2; Smith et al., 2010) with a horizontal resolution of nominal 1° (gx1v7) and 60 vertical levels. The biogeochemical component is MARBL, a prognostic ocean biogeochemistry model with a coupled cycle of carbon, macronutrients (nitrogen, phosphorus, silicate), iron, and oxygen (Long et al., 2021).

The river input of alkalinity is set as an external forcing on the estuary grids. The intensity of fluxes is based on the GlobalNEWS (Mayorga et al., 2010) and IMAGE-GNM (Beusen et al., 2015) datasets. There are two forms of inorganic carbon input into ocean module, DIC and alkalinity. Riverine DIC inputs are assumed in the form of bicarbonate (HCO<sub>3</sub>-), alkalinity flux is therefore equal to DIC influx. Readers are referred to Long et al. (2021) for more details.

The source and sink term of alkalinity in CESM include the utilization of nitrate and ammonia, the formation and dissolution of CaCO<sub>3</sub>, and release by zooplankton grazing, as shown in the following equation:

$$ALK_{tendency} = -I_{NO_3} + I_{NH_4} + 2 * CaCO_{3_{remi}} + 2 * (G - F)$$
 (1),

where the term  $I_{NO3}$  and  $I_{NH4}$  indicate the interior tendency of nitrate and ammonia,  $CaCO_{3_{remi}}$  means the dissolution of calcium carbonate, G means grazing-released alkalinity, and F means the formation of  $CaCO_3$  by small phytoplankton.

## 2.2 Experiment design

150

155

We use prognostic CO<sub>2</sub> settings to explore the responses of climate to OAE. In such a setting, dynamic atmospheric CO<sub>2</sub> forcing is used to drive the ocean and biogeochemistry module to avoid the uncertainty that stems from the difference between responsive and prescribed atmospheric CO<sub>2</sub> forcing to ocean (Tyka, 2025). The model is spun up under the pre-industrial condition (esm-piControl), which is representative of the period before the onset of large-scale industrialization with the year of 1850 as the reference year. When the climate is balanced with forcing, the historical simulation is performed as an emission-driven simulation using the historical atmospheric CO<sub>2</sub> emissions (esm-hist) prescribed by CMIP6 protocol till the year of 2014. After that, the system is forced by an emission-driven SSP5-8.5 future scenario (esmssp585; Jones et al., 2016) till 2100. Restart files in 2020 provided by the data manager of NCAR in the official CESM forum are used (https://bb.cgd.ucar.edu/cesm/). We first run a default simulation under esm-ssp585 from 2020 to 2100 as our control simulation (CTL hereafter). Then, we restart our simulation from 2020 to apply three OAE simulations and a termination of 160 OAE in 2050. The detailed simulations are as follows:

Exp1 (OWE5): 5-fold fluxes enhancement of alkalinity till 2100.

Exp2 (OWE75): A 5-fold enhancement of riverine alkalinity flux is applied from 2020 to 2049, followed by an increase to a 7.5-fold enhancement from 2050 to 2100.

Exp3 (OWE10): A 5-fold enhancement of riverine alkalinity flux is applied from 2020 to 2049, followed by an increase to a 10-fold enhancement from 2050 to 2100. 165

Exp4 (OWE0): A 5-fold enhancement of riverine alkalinity flux is applied from 2020 to 2049, followed by complete cessation of alkalinity enhancement from 2050 to 2100.

All alkalinity is added from rivers as a forcing in the model. To simplify the process, we do not change the parameters in the model but directly changed the fluxes from riverine sources. The globally integrated alkalinity fluxes are 0.11 Pmol/yr, 0.16 Pmol/yr and 0.22 Pmol/yr for OWE5, OWE75, and OWE10, which is respectively 0.088 Pmol, 0.14 Pmol, and 0.20 Pmol per year higher than the control run. Location of river mouths are shown in Fig. 1.

Figure 1: Location and fluxes of enhanced alkalinity input.

#### 3 Results 175

185

## 3.1 Responses of alkalinity to alkalinity enhancement

Compared to the control run, mean alkalinity in the upper 100 m increases throughout the simulation phase under the OWE5 treatment (Fig. 2a). By the end of 2100, it has risen by 53 meg/m³ relative to the control, demonstrating the effectiveness of alkalinity enhancement. The OWE75 and OWE10 treatments show even larger increases—77 and 102 meg/m<sup>3</sup>, respectively (Fig. 2a). Although alkalinity begins to decline after the termination of addition in 2050, the OWE0 scenario still maintains elevated alkalinity levels compared to the control run through 2100.

Globally, OAE treatments result in increased alkalinity with notable spatial variability. The largest increases occur in coastal regions and the Arctic Ocean under the OWE5 treatment, reflecting the strong influence of riverine alkalinity inputs (Fig. 3b). Beyond these areas, significant increases are observed in the Atlantic and Indian Oceans, with the Subpolar North Atlantic (SPNA) showing the highest open-ocean response. In contrast, increases in the Pacific and Southern Oceans are comparatively smaller. The spatial patterns in OWE75 are generally 190 like those of OWE5 (Fig. 3b, c). Under OWE10, however, alkalinity spreads further into subtropical regions (Fig. 3d), with particularly strong increases in the Indian Ocean and the tropical and subtropical Atlantic compared to OWE5 and OWE75. Although more modest, even the subpolar Pacific begins to show a noticeable increase under OWE10. Overall, OWE10 exhibits the strongest alkalinity enhancement of all the treatments.

While OWE0 results in a smaller alkalinity increase in the SPNA compared to other OAE 195 scenarios, this region still exhibits the most pronounced enhancement relative to the control run (Fig. 3e). The accumulation and later release of alkalinity in the Hudson Bay is a potential reason why alkalinity in the SPNA remains relatively high even after alkalinity enhancement has

ceased. During the first 30 years of OAE, alkalinity is accumulated and retained in this region due to the narrow passages in Northwestern Channel and Hudson Bay. When the OAE terminated, this accumulated alkalinity becomes a new "source", which is transported to the SPNA and effectively maintains the alkalinity compared to the control group (Fig. S1).

205

220

Vertically, alkalinity penetrates deeper in the Northern Hemisphere than in the Southern Hemisphere, likely due to two factors: (1) greater riverine alkalinity input in the north, and (2) subduction of alkalinity along with deep water formation in the high-latitude North Atlantic (Fig. 4). In all OAE treatments, the positive alkalinity anomaly rarely extends below 500 m in the Southern Hemisphere but can reach depths exceeding 1500 m around 50°–60°N (Fig. 4a–c). The polar and subpolar North Atlantic also show the sharpest vertical gradients, with alkalinity varying by more than 100 meq/m³ from the surface to 200 m in all three OAE scenarios.

The distribution of alkalinity across global ocean basins exhibits heterogeneity. In the Pacific Ocean, a positive anomaly of alkalinity is observed within the upper 300-400 m and penetrates deeper in both north and south subtropical gyres (Fig. S2a). The increase in alkalinity is greater in the north Pacific than in the south Pacific. The alkalinity anomaly in the Atlantic dominates the vertical distribution of zonal mean alkalinity anomaly, as there has the highest alkalinity increase and deepest penetration especially in SPNA as well as in subtropical gyres (Fig. S2b). In the Indian Ocean, the positive alkalinity anomaly also extends to greater depths within the subtropical gyres (Fig. S2c).

Although alkalinity addition ceases after 2050 in the OWE0 simulation, a positive alkalinity anomaly persists through 2100, reaching depths of 1500 m near 50°–70°N. However, the magnitude of this increase is much smaller compared to the continuous addition scenarios (Fig. 4d), due to the cessation of external alkalinity supply. This difference is especially noticeable in regions with strong anomalies under OWE5, such as the subpolar North Atlantic (Fig. 4e).

Figure 2: Temporal changes of (a) mean alkalinity in the upper 100 m (unit: meq/m³), (b) CO<sub>2</sub> influx (unit: mmol/m²/yr), (c) atmospheric CO<sub>2</sub> (unit: ppmv), (d) integrated DIC inventory (unit: Pmol), (e) surface pH, (f) surface air temperature (unit: °C). Perpendicular grey dash lines in the year of 2050 denote the onset of the 7.5×, 10× alkalinity enhancement scenarios (OWE 75 and OWE10, respectively), as well as the termination of alkalinity addition (OWE0). The coloured dash lines in (c), (d), (e), (f) are the anomaly between OAE simulations and the control run.

Figure 3: Distribution of alkalinity (unit: meq/m³) in upper 100 m. (a) mean alkalinity in the upper 100 m in control simulation at the end of this century, (b) differences between OWE5 and CTL, (c) differences between OWE75 and CTL, (d) differences between OWE10 and CTL, (e) differences between OWE0 and CTL. Note the different colour scale of (e). All the comparisons are based on the average of the last 10 years of simulation.

Figure 4: Vertical distribution of zonal mean alkalinity anomaly (upper 1500 m). (a) differences between OWE5 and CTL, (b) differences between OWE75 and CTL, c) differences between OWE10 and CTL, (d) differences between OWE0 and CTL, (e) differences between OWE0 and OWE5. All the comparisons are based on the average of the last 10 years of simulation.

## 3.2 CO<sub>2</sub> absorption after alkalinity enhancement

240

OAE modifies the air-sea CO<sub>2</sub> gradient, promoting greater CO<sub>2</sub> absorption in areas where the ocean is undersaturated and diminishing CO<sub>2</sub> release in regions where it is supersaturated (Fig. 5). This results in a net increase in ocean carbon storage and contributes to a reduction in atmospheric CO<sub>2</sub> levels. In all OAE scenarios, surface *p*CO<sub>2</sub> decreases significantly in coastal regions, where alkalinity is most strongly enhanced. The *p*CO<sub>2</sub> reduction also spreads into the open ocean and even reaches the Southern Ocean, despite its distance from riverine inputs (Fig. 5b–d). The magnitude of *p*CO<sub>2</sub> reduction is greater in simulations with higher alkalinity

additions, demonstrating a strong positive relationship between alkalinity enhancement and oceanic CO<sub>2</sub> uptake. In OWE5, OWE75, and OWE10, surface pCO<sub>2</sub> decreases by more than 20 µatm compared to the control, with OWE10 showing the greatest reduction. Although alkalinity addition is halted in 2050, surface  $pCO_2$  remains slightly lower by  $\sim 10$  µatm than in the control run even 50 years later (Fig. 5e). These results highlight the importance of continuous alkalinity addition to sustain enhanced CO<sub>2</sub> uptake by the ocean.

The reduction in surface pCO<sub>2</sub> drives an increase in oceanic CO<sub>2</sub> influx (Fig. 2b). Under the SSP585 scenario, the CTL exhibits a steady increase in CO<sub>2</sub> uptake until around 2080, followed by a gradual decline. In contrast, all OAE treatments show accelerated CO2 influx from the start of alkalinity addition through the end of the simulation (Fig. 2b). In OWE75 and OWE10, where alkalinity is increased further in 2050, the rate of CO<sub>2</sub> uptake accelerates even more compared to OWE5. However, a decline in CO<sub>2</sub> influx after 2080 is still observed across all OAE scenarios, although it is less pronounced than in the control run. When alkalinity addition ceases in 2050 265 (OWE0), the CO<sub>2</sub> influx rapidly returns to the same rate as in the control simulation at the 5th year after termination (Fig. 2b).

260

This enhanced oceanic CO<sub>2</sub> uptake leads to a reduction in atmospheric CO<sub>2</sub> concentrations (Fig. 2c). By 2100, the control simulation projects atmospheric CO<sub>2</sub> levels to reach 1104 ppmv. In comparison, OAE treatments lower atmospheric CO<sub>2</sub> by 29 ppmv (OWE5), 40 ppmv (OWE75), and 51 ppmv (OWE10), corresponding to reductions of approximately 2.7%, 3.6%, 270 and 4.6%, respectively. Even the OWE0 scenario results in an 11 ppmv decrease relative to the control by the end of the century.

Figure 5: Distribution of surface *p*CO<sub>2</sub>. (a) control simulation, (b) difference between OWE5 and CTL, (c) difference between OWE75 and CTL, (d) difference between OWE10 and CTL, (e) difference between OWE0 and CTL. All the comparisons are based on the average of the last 10 years of simulation.

### 3.3 Responses of DIC to alkalinity enhancement

OAE-induced CO<sub>2</sub> absorption leads to an increase in the ocean's total DIC inventory, reflecting enhanced carbon storage. In CTL, the global ocean DIC inventory continues to rise throughout the century due to the increasing atmospheric CO<sub>2</sub> partial pressure under the high-emission scenario (Fig. 2d). The OAE treatments (OWE5, OWE75, and OWE10) enable the ocean to sequester significantly more carbon than CTL during the entire OAE phase (Fig. 2d). By 2100, total ocean DIC increases by 5.39, 7.41, and 9.50 Pmol in OWE5, OWE75, and OWE10, respectively, compared to CTL. Even in OWE0, where alkalinity addition is terminated in 2050, the ocean still stores an additional 2.06 Pmol of DIC relative to the control.

Spatially, DIC inventory increases across most of the ocean in the CTL simulation due to elevated atmospheric CO<sub>2</sub> (Fig. 6a). However, the increase is not uniform—subtropical gyres, except for the North Pacific, exhibit the strongest DIC gains. In the OAE scenarios, DIC inventory increases further across nearly all ocean basins compared to CTL (Fig. 6b–d). The

largest DIC increases occur in the North Atlantic, the South Subtropical Atlantic, and the Indian Ocean, where inventories rise by more than 0.1 Tmol by 2100 under all three OAE treatments. In the North Pacific, DIC increases most notably north of the Kuroshio and its extension. As more alkalinity is added in OWE75 and OWE10, the DIC increase spreads more broadly across the oceans compared to OWE5. In contrast, the Southern Pacific exhibits only a modest increase. A slight reduction of DIC inventory is observed in the Southern Ocean under the three continuous OAE simulation relative to the control, with the intensification of this reduction under higher alkalinity addition levels (Fig. 6b-d and Fig. S3). This phenomenon is attributable to the fact that OAE effectively lowers atmospheric CO<sub>2</sub> concentrations, thereby inducing an enhanced outgassing in the Southern Ocean and ultimately leading to a net DIC inventory loss there (Fig. S4). Despite the early termination of alkalinity addition in OWE0, DIC inventory still rises across the global ocean, with the most pronounced increase occurring in the North Atlantic (Fig. 6e). However, DIC gains in OWE0 are notably smaller in the North Pacific, Indian Ocean, and South Atlantic compared to other OAE treatments.

Vertically, changes in DIC concentration mirror those of alkalinity (Figs. 7 and 4). OAE-induced 305 positive DIC anomalies reach depths of 200-500 m in the Southern Hemisphere and penetrate as deep as 700 m in the Northern Hemisphere, extending to 1500 m in the subpolar regions under all three OAE treatments (Fig. 7a-c). This pattern corresponds to strong CO<sub>2</sub> absorption in regions with elevated alkalinity (Fig. 4). The subpolar and polar regions also exhibit the steepest vertical DIC gradients, with concentrations varying by more than 100 mmol/m³ from the surface 310 to 200-300 m depth. The vertical anomaly of DIC concentration across global ocean basins generally mirror the pattern of alkalinity anomaly. The net CO<sub>2</sub> uptake induced by alkalinity injection results in DIC increase in most of ocean basins. The subtropical gyres in all three basins facilitate the downward transport of newly absorbed DIC, leading to the positive DIC anomalies in deeper layers (Fig. S5a-c). However, unlike alkalinity, a reduction in DIC concentration is 315 evident in the high-latitude regions of the Southern Hemisphere, consistent with the DIC inventory changes. Although alkalinity addition ends in 2050 in OWE0, this scenario still supports CO<sub>2</sub> uptake (Fig. 7d), with DIC increases exceeding 10 mmol/m<sup>3</sup> in the upper ocean and over 30 mmol/m<sup>3</sup> in the northern subpolar and polar regions. However, these increases are substantially smaller than those in the other OAE treatments, primarily because the decline in 320 surface alkalinity following termination reduces CO<sub>2</sub> absorption relative to OWE5 (Fig. 7e).

Figure 6: Anomaly of DIC inventory. (a) Change in DIC inventory at the end of the century relative to the initial year, (b) difference between OWE5 and CTL, (c) difference between OWE75 and CTL, (d) difference between OWE10 and CTL, and (e) difference between OWE0 and CTL. All the comparisons are based on the average of the last 10 years of simulation.

Figure 7. Zonal mean vertical distribution of DIC concentration anomalies (upper 1500 m). (a) differences between OWE5 and CTL, (b) difference between OWE75 and CTL, (c) difference between OWE10 and CTL, (d) differences between OWE0 and CTL, and (e) differences between OWE0 and OWE5. All the comparisons are based on the average of the last 10 years of simulation.

## 3.4 Responses of surface pH to alkalinity enhancement

330

OAE treatments lead to an increase in surface ocean pH, partially mitigating ocean acidification compared to the control simulation. However, they do not reverse the long-term declining pH trend throughout the 21st century (Fig. 2e). Under the high-emission SSP585 scenario, surface mean pH declines rapidly from a relatively high level (pH = 8.03) to a more acidic state (pH = 7.67) by 2100. The OWE5 treatment consistently alleviates acidification, resulting in a pH increase of 0.02 relative to the control by the end of the century. Stronger alkalinity enhancements in OWE75 and OWE10 yield even greater buffering effects, increasing surface pH

by 0.03 and 0.04, respectively. In contrast, OWE0—which terminates alkalinity addition in 2050—results in a much smaller pH increase of only 0.006 by 2100 compared to the control.

Spatially, the ocean becomes more acidic by the end of the century under the SSP585 scenario due to continued atmospheric CO<sub>2</sub> uptake (Fig. 8a). The OAE treatments mitigate acidification on a global scale, with the most substantial pH increases observed in coastal regions where riverine alkalinity input is strongest (Fig. 8b–d). In the open ocean, the Atlantic and Indian Oceans show the most notable surface pH increases, while the Southern Ocean and the Pacific exhibit more modest changes. As expected, greater alkalinity additions correspond to stronger pH buffering.

Although alkalinity input ceases in OWE0, this scenario still shows a slight increase in surface pH compared to the control, particularly in the SPNA (Fig. 8e) where there are pronounced alkalinity increase compared to the control. However, this increase is considerably smaller than those observed in the continuous OAE treatments. We also find an increase of pH in the Ross Sea by the end of this century, which is attributable to the upwelling-mediated return of OAE-induced alkalinity to the surface, thereby elevating surface pH (Fig. S6).

Figure 8: Distribution of surface pH. (a) control simulation at the end of the century, (b) difference between OWE5 and CTL, (c) difference between OWE75 and CTL, (d) difference between OWE10 and CTL, (e) difference between OWE0 and CTL. Note the

different colour scale in (e). All the comparisons are based on the average of the last 10 years of simulation.

## 3.5 Responses of surface air temperature to alkalinity enhancement

To figure out how OAE can mitigate global warming, we calculate the surface air temperature in all simulations. There is a continuous increase in temperature under CTL, indicating that the temperature keeps rising under the high-emission scenario (Fig. 2f). All the four OAE treatments show a slight decrease of temperature in the last 10 years of this century, with 0.45 °C in OWE5, 0.39 °C in OWE75, 0.34 °C in OWE10, and 0.31 °C in OWE0 compared to CTL (Fig. 2f).

#### 4 Discussion

380

390

395

### 4.1 The effectiveness of OAE via rivers

The long-standing negative feedback between terrestrial weathering and the oceanic carbon sink—mediated through riverine alkalinity input—makes river-based OAE a practical and feasible carbon removal strategy. In our simulations, we apply 0.088, 0.14, and 0.20 Pmol of excess alkalinity per year at river mouths, representing 5-, 7.5-, and 10-fold increases in alkalinity fluxes due to enhanced weathering. These treatments lead to reductions in atmospheric CO<sub>2</sub> of 29, 40, and 51 ppm, respectively, which are consistent with the range reported in previous modeling studies. For example, Ilyina et al. (2013) show that doubling the amount of alkalinity relative to CO<sub>2</sub> emissions could lower atmospheric CO<sub>2</sub> by up to 490 ppm. Similarly, González and Ilyina (2016) demonstrate that adding 114 Pmol of alkalinity to the surface ocean under the RCP8.5 scenario could stabilize atmospheric CO<sub>2</sub> at RCP4.5 levels (~520 ppm). While the magnitude of alkalinity addition in these studies is equivalent to 100 times the present-day weathering rate—ten times higher than in our OWE10 scenario—the resulting atmospheric CO<sub>2</sub> reductions are also nearly tenfold greater. In our simulation, a total of 12.64 Pmol of alkalinity added over 80 years yields a 51 ppm CO<sub>2</sub> reduction. Likewise, an alkalinity addition of 0.25 Pmol yr<sup>-1</sup> lead to an 82-86 ppm decline in atmospheric CO<sub>2</sub> under a high emission scenario, and a 53-58 ppm decline under a low emission scenario (Lenton et al., 2018). Our OWE75 simulation, which adds 0.14 Pmol yr<sup>-1</sup> via rivers, achieved a 40 ppm reduction, roughly half the CO<sub>2</sub> drawdown observed in Lenton et al. (2018), in line with the halved addition rate. These comparisons suggest a robust linear relationship between the amount of alkalinity added and the resulting atmospheric CO<sub>2</sub> reduction. However, further research is needed to confirm the consistency and limits of this relationship. Furthermore, Schwinger et al. (2024) suggest that the efficiency of CO<sub>2</sub> drawdown via OAE is influenced by the emission pathway, with greater declines under high-emission scenarios compared to low-emission ones. Therefore, under loweremission scenarios, the CO<sub>2</sub> reduction achieved through riverine OAE is likely to be less pronounced than what we observe under the SSP585 scenario.

We calculate additional DIC stored per unit of TA added, referred to as absorption efficiency, following the definition by Palmiéri and Yool (2024). By the end of the century, the OAE

simulations result in DIC inventory increases of 5.39, 7.41, and 9.50 Pmol in the OWE5, OWE75, and OWE10 simulations, respectively. These correspond to absorption efficiencies of 0.77, 0.77, and 0.75. Our results fall within the range reported in previous studies (0.46 - 0.95). 400 The wide range in previous studies is probably due to spatial and temporal variability, as well as differences in OAE application methods, which will influence the contact time between the water mass and the atmosphere and the time for water mass to reach equilibrium. Our results are consistent with Palmiéri and Yool (2024), who reported a global mean absorption efficiency of 0.78 when OAE is applied in shallow continental shelves. In our simulations, the Southern 405 Ocean exhibits the smallest increase in DIC due to its distance from the riverine alkalinity sources. However, when OAE is applied specifically in the Southern Ocean, higher absorption efficiencies have been reported compared to uniform global deployment (Burt et al., 2021). This suggests that the Southern Ocean may be a particularly effective region for OAE, but further studies are needed to confirm its potential. 410

Most ESMs do not take into account sediment processes, or they treat sediment processes as a part of the closed calcium carbonate cycle without considering the complex processes of sedimentation (Planchat et al., 2023). The absence of sedimentation processes may lead to an overestimation of the efficiency of OAE on a longer time scale. Using a simple carbon cycle box-model, Köhler (2020) showed that the absorption efficiency gradually increases to a peak 415 value of 0.81 at the time of maximum atmospheric CO<sub>2</sub> emissions, then declines to half of that peak after a 2000-year simulation, due to the deepened calcite saturation horizon and lysocline transition zones in sediment, which will lead to an increase of CaCO3 accumulation. In a relative short-term simulation, a gradual increase in absorption efficiency is observed when alkalinity is 420 added in coastal regions (He & Tyka, 2023). In our three OAE simulations (OWE5, OWE75, and OWE10), absorption efficiency increases from 0.2 in the first year to 0.77, 0.77, and 0.75 by the end of the century, respectively, which are consistent with those of He and Tyka (2023). Although the short simulation in He and Tyka (2023) and our study likely missed the decline stage in adsorption efficiency in Köhler (2020), but the lack of sediment processes will overrate the efficiency later than 2100. Additionally, we find that absorption efficiency is slightly lower in 425 simulations where more alkalinity is added. For example, the efficiency in OWE10 is 0.75, compared to 0.77 in both OWE5 and OWE75, indicating a trade-off between the amount of alkalinity added and the resulting efficiency. In practice, it is essential to identify optimal deployment strategies that maximize efficiency while minimizing cost.

### 430 **4.2 OAE distribution affected by circulations**

Although alkalinity is introduced via rivers, its effects extend to the open oceans, with more pronounced impacts observed in the Atlantic and Indian Oceans compared to the Pacific (Fig. 3). This pattern is likely driven by differences in ocean topography and circulation. For instance, in the Atlantic, excess alkalinity from the Caribbean Sea can be transported to the North Atlantic by the Gulf Stream, a strong western boundary current. Compared to the North Atlantic, the western boundary current of the North Pacific occurs outside the island chains, and a large amount of

ALK excess is enriched inside the island chains, preventing it from spreading to the wider Pacific. Moreover, Zhou et al. (2024) reported that absorption efficiency is higher when OAE is applied in the equatorial Pacific than in subtropical regions. In contrast, our simulations show low absorption efficiency in the equatorial Pacific and only minimal increases in DIC inventory. We attribute this discrepancy to differences in the calculation methods. Zhou et al. (2024) applied OAE regionally, adding alkalinity to all grid cells within selected regions, and defined efficiency as the ratio of the global increase in DIC inventory to the total alkalinity added. Their finding of high efficiency in the equatorial Pacific is intuitive, as upwelling there spreads 445 additional alkalinity across the surface ocean, enhancing CO<sub>2</sub> uptake. By contrast, in the subtropical gyres, which is characterized by convergence, added alkalinity is more readily subducted into the deep ocean, reducing efficiency. In our approach, however, efficiency is calculated locally as the ratio of the increase in DIC inventory to the increase in alkalinity within the same region when regional efficiency is calculated. Under this definition, strong upwelling in the equatorial Pacific promotes CO<sub>2</sub> outgassing, resulting in lower local efficiency. 450

The vertical distribution of excess alkalinity further illustrates the influence of meridional circulation on OAE. The most affected region is the SPNA, where excess alkalinity penetrates to depths greater than 1500 m due to deep-water formation (Fig. 4a). In both the north and south subtropical gyres, the downward sloping of isoclines indicates a convergent effect that facilitates the subduction of alkalinity. In equatorial regions, the positive alkalinity anomaly remains shallower than in the subtropical gyres, suggesting that upwelling in these regions retains excess alkalinity near the surface. This pattern aligns with the findings of Lenton et al. (2018). It is important to note that OAE occurring in deep-water formation or convergence regions does not necessarily imply lower efficiency. Nagwekar et al. (2024) demonstrate through modeling that mCDR in these regions remains effective and holds significant potential.

## 4.3 Challenges of OAE

One of the most critical challenges in OAE is alkalinity loss through precipitation, which can rapidly reduce efficiency (Moras et al., 2022). The extent of this loss depends on the type and form of added material, solution state, and presence of particles (Hartmann et al., 2023). For riverine OAE, substantial losses may occur in estuaries, making it essential to regulate addition rates. CO<sub>2</sub>-equilibrated alkaline solutions and certain Mg-rich minerals can help limit precipitation (Jones, 2017; Pan et al., 2021), though some, like olivine, may still be less efficient due to particle-induced losses (Fuhr et al., 2022). Using finely ground particles can improve dissolution but increases energy costs, while particles in river plumes can promote heterogeneous precipitation (Wurgaft et al., 2021). These factors highlight the need for careful material selection and delivery design to minimize losses in real-world applications.

Meeting the substantial material requirements for OAE materials presents another major challenge. To achieve annual OAE rate of 0.088, 0.14 and 0.20 Pmol, and assuming that each

mole of olivine releases 4 moles of alkalinity with a mole mass of 140 g mol<sup>-1</sup> (Feng et al., 2017), approximately 3.08, 4.9, and 7 billion tons of olivine would be required per year, respectively. The current global production of olivine at only 8.4 million tons annually - far below the required amount (Caserini et al., 2022). Industrial by-products such as slag, cement, and lime could partially offset this shortfall. These materials collectively represent around 7 billion tons of alkaline output annually and have a CO<sub>2</sub> storage potential of 2.9-8.5 billion tons per year (Renforth, 2019). Thus, they offer a promising supplemental source of alkalinity for OAE. Restoration of blue carbon ecosystems may also contribute additional alkalinity. For instance, mangrove restoration can enhance respiration and sediment dissolution, increasing local alkalinity levels (Fakhraee et al., 2023). However, the overall magnitude and long-term effectiveness of this approach remain uncertain and warrant further investigation.

Several additional concerns deserve attention. Bach (2024) raised the issue of "additionality", 485 where anthropogenic alkalinity inputs may reduce the dissolution of natural alkalinity sources. This highlights the need for further study of interactions between natural and artificial alkalinity sources under OAE scenarios. Furthermore, González et al. (2018) found that warming and acidification rates can accelerate following the cessation of alkalinity enhancement, suggesting that long-term deployment strategies are necessary to avoid abrupt climate rebound effects. Our 490 own simulations also show a rapid decline in pH and CO<sub>2</sub> uptake after alkalinity addition ceases, underscoring the importance of sustained application. Moreover, the increase in ocean carbon uptake is partially offset by a corresponding decrease in the land carbon sink of -4.31, -7.05 and -9.20 PgC in the OWE5, OWE75 and OWE10 simulations, respectively. These results underscore the importance of considering terrestrial carbon dynamics when evaluating the net effectiveness 495 of OAE. To avoid offsetting the benefits of OAE, complementary strategies to preserve or enhance land carbon sequestration may be necessary.

Importantly, OAE at the levels of alkalinity addition used in our study is insufficient to reverse the trajectory of climate change. Under the SSP585 scenario, global temperature keeps increasing even with OAE interventions. While OAE treatments slightly reduce surface temperature, they fail to halt the overall warming trend. These results are consistent with Lenton et al. (2018), who found that uniform global alkalinity addition under the RCP8.5 scenario produced only modest climate effects. Therefore, in high-emission scenarios, increasing ocean alkalinity may help mitigate ocean acidification but is unlikely to substantially reduce global temperatures. Additional carbon dioxide removal (CDR) strategies, such as solar radiation management (SRM), may be required to avoid dangerous temperature overshoot. We also find the reductions in surface air temperature are not proportional to alkalinity addition. This is because the slight cooling induced by OAE is smaller than the interannual variability simulated by the model, and is therefore obscured by internal climate variability (Lenton et al., 2018). We believe this phenomenon warrants further investigation with larger ensembles or longer simulations to confirm its robustness.

500

## 5 Conclusions and unresolved problems

In this study, we evaluated the efficacy of ocean alkalinity enhancement (OAE) using the CESM-MARBL model under a high-emission scenario, employing an idealized method of alkalinity addition. Although the model's spatial resolution is relatively coarse for capturing the complex physical processes in coastal regions, the riverine source and sink terms have been extensively calibrated against observational data. Furthermore, our simulations do not alter the model's physical dynamics; only the magnitude of alkalinity inputs is modified. Therefore, we believe that the impact of model resolution on our results is minimal. To focus on the theoretical potential of OAE, we simulated the addition of "pure alkalinity." However, real-world applications must contend with challenges such as alkalinity precipitation, particularly runaway precipitation. Future OAE modeling efforts should incorporate additional constraints and empirical formulations to better represent the behavior of actual alkaline particles (Fennel et al., 2023).

Our findings show that even a tenfold increase in alkalinity flux is insufficient to reverse the trends of ocean acidification or rising atmospheric CO<sub>2</sub> levels under a high-emission scenario. This underscores the urgent need for complementary technological innovations and aggressive emission reduction strategies to curb CO<sub>2</sub> emissions. In parallel, other CDR techniques will be necessary to actively draw down atmospheric CO<sub>2</sub>. Additionally, in our OWE0 simulation, if alkalinity addition ceases in 2050, the ocean continues to absorb atmospheric CO<sub>2</sub>, though at a reduced rate compared to scenarios where alkalinity addition persists through the end of the century. This indicates that water masses altered by OAE and not in equilibrium with the atmosphere will return to the surface through ocean circulation on centennial timescales.

#### **Author Contributions**

W.-L. W. conceived the project. X.-Y. Z carried out the formal analyses with inputs from W.-L. W and S. L. X.-Y. Z., W.-L. W., and S. L. wrote the manuscript. All authors have given approval to the final version of the manuscript.

## Code/Data availability

There is no new code created in this study. Data are available at the following repository https://doi.org/10.5281/zenodo.15550663

## **Competing interests**

The authors declare that there is no competing interests.

## Acknowledgments

We thank the hundreds of scientists and researchers who built and has been maintaining the Community Earth System Model. W.-L.W and X.-Y.Z. were supported by the National Natural Science Foundation of China (42476031), the National Key Research and Development Program of China (NKPs) 2023YFF0805004 and the Natural Science Foundation of Fujian Province of China 2023J02001.

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
