# Peer review of "Assessing the efficacy of river-based ocean alkalinity enhancement for carbon sequestration under high emission pathways"

_EGUsphere, 2025_

## Author Response (AR1)

**Responses to editor:**

We sincerely thank the editor and reviewer for the thorough evaluation of our manuscript and constructive suggestions provided. We have carefully considered each comment and made corresponding revisions to improve the clarity, accuracy, and overall quality of the work. Below, we provide a detailed, point-by-point response to all comments.

The reviewers find the study conceptually strong but agree that substantial revisions are required. Both emphasize that while you provided useful clarifications in your responses, these must be explicitly integrated into the manuscript. Key issues include the need for a clearer discussion of atmospheric feedbacks and temperature responses as well as the omission of how the land carbon sink responds to OAE. In addition, the figures and presentation require revision to improve clarity, and the novelty of the river-based OAE framing should be articulated more strongly. Please address these points thoroughly in your revision and demonstrate how changes were incorporated. The revised manuscript will be sent back to the reviewers for reassessment.

We thank you for the careful review of our manuscript and the valuable comments and suggestions that you have provided. Your professional insights have been instrumental in helping us improve the quality and rigor of the paper, and we greatly appreciate the time and effort you have dedicated to this process.

Key issues include the need for a clearer discussion of atmospheric feedbacks and temperature responses as well as the omission of how the land carbon sink responds to OAE.

Indeed, both reviewers raised similar concerns. We have addressed these points in detail in our reply letter and provide a summary of the questions and our responses below.

Reviewer #2, for example, highlighted the following issue:

"269-282: The DIC decrease in the Southern Ocean and equatorial Pacific is not mentioned? This is an apparent phenomenon, and overlooking it is weird. This may be due to the atmospheric feedback effects present in the ESM, for example, the ALK injection in the northern oceans reducing PCO2atm, which could have led to net outgassing in the Southern Ocean. However, further analysis is needed to confirm this. A more detailed analysis of the atmospheric and surface ocean PCO2 outputs for both the CTRL and experimental groups is necessary to determine whether the DIC decrease is due to atmospheric feedback or other mechanisms."

Our analysis shows that in the Southern Ocean, the difference between atmospheric and oceanic surface ocean $p\mathrm{CO_2}$ is smaller in the OAE simulations than in the control run (Fig. S2), indicating enhanced outgassing under OAE. This mechanism likely explains the observed DIC decrease and becomes more pronounced with higher levels of alkalinity addition (Fig. S3), consistent with the reviewer's expectation.

However, this explanation applies only where the atmospheric $CO_2$ decrease exceeds the corresponding seawater $pCO_2$ decrease. In the equatorial Pacific, the OAE-induced reduction in seawater $pCO_2$ is comparable to that in atmospheric $CO_2$, resulting in only a slight reduction in outgassing and a small net increase in DIC inventory. We have added these clarifications to Section 3.3 of the revised manuscript in line 296-302.

*"A slight reduction of DIC inventory is observed in the Southern Ocean under the three continuous OAE simulation relative to the control, with the intensification of this reduction under higher alkalinity addition levels (Fig. 6b-d and Fig. S3). This phenomenon is attributable to the fact that OAE effectively lowers atmospheric $CO_2$ concentrations, thereby inducing an enhanced outgassing in the Southern Ocean and ultimately leading to a net DIC inventory loss there (Fig. S4)."*

Reviewer #1 raised the following concerns on the temperature responses:

L334-335 Does this mean the reductions in atmospheric air temperatures are not proportional to OAE? This is an important finding and requires discussion which appears to be absent. Why do the authors think this is the case? Is this because of internal variability? Are larger ensemble sizes of each experiment required?

We think these disproportional reductions of temperature are relative to the smaller atmospheric $CO_2$ declines which only lead to a 10% decrease of temperature compared to the temperature increase under esm-SSP585 scenario. Thus, the interannual temperature cover up the temperature decrease induced by riverine OAE. We have added these discussions in our discussion section in line 507-512.

*"We also find the reductions in surface air temperature are not proportional to alkalinity addition. This is because the slight cooling induced by OAE is smaller than the interannual variability simulated by the model, and is therefore obscured by internal climate variability (Lenton et al., 2018). We believe this phenomenon warrants further investigation with larger ensembles or longer simulations to confirm its robustness."*

Reviewer #1 also asked how the land carbon sink responds to OAE. To answer this question, we have integrated the total column carbon in land as the land carbon sink. We find the land carbon sink decreases by 4.31, 7.05 and 9.20 PgC in the simulation of OWE5, OWE75 and OWE10 respectively. We have added these results in discussion section in line 493-498.

*"Moreover, the increase in ocean carbon uptake is partially offset by a corresponding decrease in the land carbon sink of -4.31, -7.05 and -9.20 PgC in the OWE5, OWE75*

*and OWE10 simulations, respectively. These results underscore the importance of considering terrestrial carbon dynamics when evaluating the net effectiveness of OAE. To avoid offsetting the benefits of OAE, complementary strategies to preserve or enhance land carbon sequestration may be necessary."*

In addition, the figures and presentation require revision to improve clarity, and the novelty of the river-based OAE framing should be articulated more strongly.

We have revised all the figures, enlarged the size of the legends to make the figures clearer.

The revised manuscript and responses to all reviewers are attached to this letter. We believe these revisions will enhance the overall quality of the manuscript.

**Response to Reviewer #1**

We sincerely thank the reviewer for their insightful and constructive comments. We appreciate the acknowledgment that expanding the diversity of OAE simulation studies is important. Below, we provide a point-by-point response to the main concerns raised.

**General Comments**

The following article addresses the impact of river-focused ocean alkalinity enhancement on carbon dioxide removal. It present's findings that mCDR broadly scales with OAE as other studies have similarly shown. While I believe that it's important to expand the number of OAE simulation studies and varying the means of alkalinity delivery is critical, the article is not particularly interesting. The authors could do more to differentiate their contribution, particularly given their use of an emissions-driven ESM. I was particularly surprised that they focus so little on changes in atmospheric temperatures, which appear counterintuitive. Moreover, there is no description at all of the land carbon sink and how it responds to OAE (one of the principal advantages of using a fully-coupled ESM). I would like to see both of these aspects developed in a revised manuscript. In my opinion, several of the current figures need cutting or revising to be useful to the reader.

**In the following, we address the general comments individually, providing responses to each point.**

**Reviewer Comment:** "The article is not particularly interesting. The authors could do more to differentiate their contribution, particularly given their use of an emissions-driven ESM."

**Response:**
We appreciate the suggestion and have revised the manuscript to better emphasize the novelty of our work. Specifically, our study:

- Implements *river-based* alkalinity enhancement, reflecting a natural and spatially realistic pathway of alkalinity delivery that differs from the commonly assumed uniform ocean-wide input.
- Uses an *emissions-driven*, fully coupled Earth System Model (CESM2), which allows for two-way interactions between climate, ocean chemistry, and carbon fluxes—features not captured in prescribed-$CO_2$ simulations.
- Explores *termination effects* of OAE (OWE0) in addition to scaling scenarios, providing insights into persistence and reversibility of OAE-induced changes.

We have added more discussion on the temporal changes of air temperature and land carbon sink. We also reemphasize the novelty and highlight the contributions of the current work in the revised manuscript and also as follows.

In line 108-115, we emphasize the novelty of our study:

*"…To mimic this mechanism, we use an emission-driven, fully coupled Earth System*
*Model to evaluate a riverine-based, global-scale OAE scenario under a high-emission*
*pathway (Shared Socioeconomic Pathway 5-8.5, SSP585), which reflects a natural*
*and spatially realistic pathway of alkalinity delivery that differs from the commonly*
*assumed uniform ocean-wide input. The responsive $CO_2$ concentration configuration*
*in our simulation allows the interactions between climate, ocean chemistry and*
*carbon fluxes and captures the features not captured in prescribed-$CO_2$ simulations."*

In line 120-123, we highlight the contributions of this work:

*"This study provides the transient responses of ocean system to OAE and insights into*
*persistence and reversibility of OAE-induced changes, as well as the suggestion to*
*future study and deployment of OAE."*

In line 507-512, we discuss the changes of air temperature:

*"…We also find that reductions in surface air temperature are not proportional to the*
*level of alkalinity addition. This is because the slight cooling induced by OAE is*
*smaller than the interannual variability simulated by the model, and is therefore*
*obscured by internal climate variability (Lenton et al., 2018). We believe this*
*phenomenon warrants further investigation with larger ensembles or longer*
*simulations to confirm its robustness."*

In line 493-498, we calculate and discuss the land carbon sink:

*"Moreover, the increase in ocean carbon uptake is partially offset by a corresponding*
*decrease in the land carbon sink of –4.31, –7.05, and –9.20 PgC in the OWE5,*
*OWE75, and OWE10 simulations, respectively. These results underscore the*
*importance of considering terrestrial carbon dynamics when evaluating the net*
*effectiveness of ocean alkalinity enhancement. To avoid offsetting the benefits of OAE,*
*complementary strategies to preserve or enhance land carbon sequestration may be*
*necessary."*

**Reviewer Comment:** "I was particularly surprised that they focus so little on changes
in atmospheric temperatures, which appear counterintuitive."

**Response:**

We thank the reviewer for highlighting this important and counterintuitive aspect of
our results. In the revised manuscript, we have expanded the Discussion section to
address the changes in atmospheric temperature under riverine OAE scenarios.

Our results show that reductions in surface air temperature are not proportional to the amount of alkalinity added. This disproportionality is primarily due to the relatively modest declines in atmospheric $CO_2$, which lead to only a ~10% decrease in temperature relative to the projected warming under the baseline esm-SSP585 scenario. As a result, the temperature reductions associated with OAE are small and largely masked by interannual variability in the Earth system model.

We have added the discussion about surface air temperature change in line 507-512

*"...We also find that reductions in surface air temperature are not proportional to the level of alkalinity addition. This is because the slight cooling induced by OAE is smaller than the interannual variability simulated by the model, and is therefore obscured by internal climate variability (Lenton et al., 2018). We believe this phenomenon warrants further investigation with larger ensembles or longer simulations to confirm its robustness."*

**Reviewer Comment:** "There is no description at all of the land carbon sink and how it responds to OAE (one of the principal advantages of using a fully-coupled ESM)."

**Response:**

We appreciate this insightful comment. In response, we have added more discussion in the revised manuscript that quantifies changes in the terrestrial carbon sink under each OAE scenario.

To evaluate the land carbon sink, we calculated the total column-integrated carbon over land areas. Our results show that the land carbon sink declines by 4.31, 7.05, and 9.20 PgC in the OWE5, OWE75, and OWE10 simulations, respectively. These values have also been incorporated into the revised Discussion section in line 493-498, where we state:

*"...Moreover, the increase in ocean carbon uptake is partially offset by a corresponding decrease in the land carbon sink of −4.31, −7.05, and −9.20 PgC in the OWE5, OWE75, and OWE10 simulations, respectively. These results underscore the importance of considering terrestrial carbon dynamics when evaluating the net effectiveness of ocean alkalinity enhancement. To avoid offsetting the benefits of OAE, complementary strategies to preserve or enhance land carbon sequestration may be necessary."*

**Reviewer Comment:** "Several of the current figures need cutting or revising to be useful to the reader."

**Response:**

Thank you for this helpful suggestion. In response:

- We have removed Figure 1.
- We have redrawn Figure 3 to make it clear.
- Font sizes and color schemes have been adjusted throughout for better readability for figure 5.

We hope that these revisions improve the manuscript's readability and impact.

Specific Comments:

L26 Is this true? Wouldn't afforestation-based mCDR also absorb $CO_2$ and reduce acidification?

We agree with the reviewer that afforestation-based mCDR can also contribute to $CO_2$ removal and, indirectly, to the mitigation of ocean acidification. Afforestation enhances atmospheric $CO_2$ uptake through biological carbon sequestration, which in turn reduces the partial pressure of $CO_2$ in surface waters, thereby decreasing $CO_2$ dissolution and alleviating acidification (N'Yeurt et al., 2012). In contrast, OAE reduces acidification more directly by adding alkaline substances that chemically neutralize $H^+$ ions in seawater. To reflect this distinction and avoid overstating the uniqueness of OAE, we have revised the sentence as follows in line 26-27:

*"...is one of the promising Carbon Dioxide Removal methods that can simultaneously absorb $CO_2$ and alleviate ocean acidification."*

L34-35 These are surface atmospheric temperature increases not SST increases I believe.

We are grateful to the reviewer for pointing out the mistake. We have revised this sentence accordingly as follows in line 34-37:

*"...Global average surface atmospheric temperature has already increased by 1.1 °C relative to the 1850–1900 baseline (IPCC, 2023) and continues to rise, approaching the Paris Agreement's target of limiting warming to below 1.5 °C by the end of this century (UNFCCC, 2015)."*

L53 I would use a more recent estimate of this consistent with the latest scenarios (e.g. (Smith et al., 2024))

Thank you. The numbers have been updated according to estimate by Smith et al. (2024) in line 50-52:

*"...However, an additional $CO_2$ sequestration requirement of -5.3 $GtCO_2$ per year is needed on the base of -2.1 $GtCO_2$ per year in 2011-2020 even under 76% greenhouse gas emission reduction (Smith et al., 2024)."*

L59 Excluding geological reservoirs.

We sincerely thank you for pointing out the inaccurate expression in our manuscript. We have revised this sentence and describe the ocean as the largest carbon reservoir on Earth surface in line 57-59.

*"…As the largest carbon reservoir at the Earth's surface, the ocean holds substantial potential for enhanced $CO_2$ uptake."*

L65-67 See previous point, other techniques could potentially also do this.

Agreed. We have adjusted the tone accordingly, both in our response and in the revised manuscript in line 64-66:

*"…Among these, OAE is promising because it offers the dual benefit of reducing atmospheric $CO_2$ and direct effect on alleviating ocean acidification, making it an ideal candidate for mitigating $CO_2$-driven climate impacts through mCDR."*

L68-70 This definition is a bit inaccurate. Alkalinity is perhaps better defined as the excess of $H^+$ accepters over donors.

Agreed. We have modified the definition accordingly, both in our response and in the revised manuscript in line 67.

*"Alkalinity is defined as the excess of proton acceptors over proton donators in seawater."*

L70-71 This alkalinity decline may also be due to biotic feedbacks, (Barrett et al., 2025; Kwiatkowski et al., 2025).

Agreed. we have modified the sentence accordingly as follows and also in the revised manuscript in line 69-71:

*"…A decline in surface alkalinity, driven by enhanced upper-ocean stratification and bio-activity, has been shown to reduce oceanic carbon uptake (Barrett et al., 2025; Kwiatkowski et al., 2025)."*

L73 I'm not sure what excess $H^+$ is in this context.

We appreciate the reviewer's comment. Our original intention was to describe the removal of additional protons resulting from ocean acidification. However, we agree that the term "excess $H^+$" is potentially misleading and redundant with the accompanying description of rising pH. Therefore, we have removed this phrase in the revised manuscript.

In line 71-74, the revised sentence now reads:

*"OAE works by introducing carbonate, bicarbonate, or other H⁺ acceptors into surface waters, thereby increasing carbonate ion concentrations, raising pH, and reducing the partial pressure of $CO_2$ ($pCO_2$) in seawater."*

L74-75 Disequilibrium is not always enhanced. In areas of natural carbon outgassing, such as eastern boundary upwelling systems, it would likely be reduced. The net effect would be the same however, enhanced ocean carbon storage.

We thank the reviewer for this helpful clarification. We agree that air–sea $CO_2$ disequilibrium is not uniformly enhanced across all regions, particularly in natural outgassing areas such as eastern boundary upwelling systems, where disequilibrium may actually be reduced. To improve clarity, we have revised the sentence accordingly in line 74-76:

*"By altering the air–sea $CO_2$ disequilibrium, OAE can enhance oceanic $CO_2$ uptake in undersaturated regions and reduce outgassing in oversaturated regions, thereby increasing net ocean carbon storage and ultimately lowering atmospheric $CO_2$ concentrations."*

L103-104 There are a growing number of regional OAE simulation studies that go beyond this, some of which the authors go on to cite.

Thank you for your suggestion. Now we have modified this sentence as follows in line 103-105:

*"...Although there are a growing number of regional OAE simulations in recent years (e.g. Burt et al., 2021; Feng et al., 2017; He & Tyka, 2023), we still lack research using more practical delivery methods, such as river-based OAE."*

Figure 1. I don't find this figure particularly useful. The link between weathering and atmospheric $CO_2$ is unclear to me. Is this due to intensification of the hydrological cycle? And the role of sources and sinks of alkalinity in ocean sediments and marine biota is absent.

We have removed this figure.

L130 This equation is unnecessary (and is unnumbered).

Agreed. We have removed this equation.

L141 Add equation number.

Added.

L145-149. The language used here is not clear. Prescribed $CO_2$ can still be transiently changing. Are simulations concentration-driven or emissions-driven? If emissions-driven, with dynamic atmospheric $CO_2$ this needs to be explicit here.

We agree that in a prescribed $CO_2$ configuration $CO_2$ concentration will transiently change in atmosphere module. In such a setting, the atmospheric $CO_2$ forcing driving the ocean module changes in a fixed trajectory. Whereas, in a prognostic $CO_2$ configuration, atmospheric $CO_2$ concentration is dynamically changed according to the net strength of sources (e.g., emission) and sinks (e.g., land and ocean sinks). We expended the clarification of prognostic $CO_2$ in line 148-151:

*"...We use prognostic $CO_2$ settings to explore the responses of climate to OAE. In such a setting, dynamic atmospheric $CO_2$ forcing is used to drive the ocean and biogeochemistry module to avoid the uncertainty that stems from the difference between responsive and prescribed atmospheric $CO_2$ forcing to ocean (Tyka, 2025)."*

L153 "concentration" should be "emissions" as emissions not concentrations are prescribed in esm-hist.

Thank you for pointing this out. We have changed the "concentration" to "emissions". In line 153-156:

*"When the climate is balanced with forcing, the historical simulation is performed as an emission-driven simulation using the historical atmospheric $CO_2$ emissions (esm-hist) prescribed by CMIP6 protocol till the year of 2014."*

L155 I don't know what an SSP-based RCP is. You either ran an SSP or an RCP or is this some hybrid forcing I am not aware of.

Thank you for catching this mistake. We indeed used the emissions-driven SSP5-8.5 forcing scenario (esm-ssp585), not the concentration-driven variant. We have corrected the sentence in the revised manuscript to reflect this accurately. In line 156-157:

*"After that, the system is forced by an emission-driven SSP5-8.5 future scenario (esm-ssp585; Jones et al., 2016) till 2100."*

L162-164 These simulation descriptions are confusing. What is meant by "based on… from 2050"?

In our simulation setup, we first ran the OWE5 scenario continuously from 2020 to 2100. Based on the conditions and outputs from the first 30 years of the OWE5 simulation, we then initialized three additional scenarios—OWE75, OWE10, and OWE0—starting from the year 2050 and continuing through 2100. In these latter simulations, the riverine alkalinity flux was modified relative to OWE5 beginning in year 2050, corresponding to year 30 of the OWE5 run. In line 162-168:

*"...*

*Exp2 (OWE75): A 5-fold enhancement of riverine alkalinity flux is applied from 2020 to 2049, followed by an increase to a 7.5-fold enhancement from 2050 to 2100.*

*Exp3 (OWE10): A 5-fold enhancement of riverine alkalinity flux is applied from 2020 to 2049, followed by an increase to a 10-fold enhancement from 2050 to 2100.*

*Exp4 (OWE0): A 5-fold enhancement of riverine alkalinity flux is applied from 2020 to 2049, followed by complete cessation of alkalinity enhancement from 2050 to 2100."*

L165-169 Is the ocean alkalinity inventory balanced in the control run? Or is there some drift?

The model was spun-up by the community. In the spin-up runs, the burial of $CaCO_3$ was tuned to balance the alkalinity input from rivers (Long et al., 2021). We do not rerun the spin-up phase and used the the restart files of the year 2020 obtained from data manager in CESM forum (https://bb.cgd.ucar.edu/cesm/). There may be still a trivial drift in these restart files, but it should not have a significant impact on our simulation because we conducted the control run and OAE simulations using the same restart file. Therefore, any drift should be canceled.

Figure 3 In printed format it is impossible to see any of the detail of this figure. Fonts are too small, lines to thin and legends impossible to read.

The image quality might be compressed when generating the PDF. In any case, we have redrawn the figure and made it clear. And now it is Fig.2.

"...

[Figure]

Figure 2: Temporal changes of (a) mean alkalinity in the upper 100 m (unit: meq/m³), (b) CO₂ influx
(unit: mmol/m²/yr), (c) atmospheric CO₂ (unit: ppmv), (d) integrated DIC inventory (unit: Pmol),
(e) surface pH, (f) surface air temperature (unit: °C). Perpendicular grey dash lines in the year of
denote the onset of the 7.5×, 10× alkalinity enhancement scenarios (OWE 75 and OWE10,
respectively), as well as the termination of alkalinity addition (OWE0). The coloured dash lines in
(c), (d), (e), (f) are the anomaly between OAE simulations and the control run."

L220 Clarify in the legend whether these are global zonal means or a specific transect.

Thank you. It is zonal mean. We have modified the legend of the Fig. 4 in our revised manuscript
as follows:

"...Figure 4: Vertical distribution of zonal mean alkalinity anomaly (upper 1500 m). (a) differences

*between OWE5 and CTL, (b) differences between OWE75 and CTL, c) differences between OWE10*
*and CTL, (d) differences between OWE0 and CTL, (e) differences between OWE0 and OWE5. All*
*the comparisons are based on the average of the last 10 years of simulation."*

L225-226 See earlier point. OAE does not always enhance disequilibrium. If it does, I would like
to see a plot of this.

We now have modified this part as follows in line 246-249:

*"...OAE modifies the air–sea $CO_2$ gradient, promoting greater $CO_2$ absorption in areas where the*
*ocean is undersaturated and diminishing $CO_2$ release in regions where it is supersaturated. This*
*results in a net increase in ocean carbon storage and contributes to a reduction in atmospheric $CO_2$*
*levels."*

L235 I think uatm units should be used for partial pressures.

Thank you for your suggestion. We have changed the units to uatm in line 254-257 and in Fig. 6:

*"In OWE5, OWE75, and OWE10, surface $pCO_2$ decreases by more than 20 μatm compared to the*
*control, with OWE10 showing the greatest reduction. Although alkalinity addition is halted in 2050,*
*surface $pCO_2$ remains slightly lower by ~10 μatm than in the control run even 50 years later*
*(Fig. 5e)."*

"

[Figure]

*Figure 5: Distribution of surface $pCO_2$. (a) control simulation, (b) difference between OWE5 and CTL, (c) difference between OWE75 and CTL, (d) difference between OWE10 and CTL, (e) difference between OWE0 and CTL. All the comparisons are based on the average of the last 10 years of simulation.*

*"*

L245 How much later? Give the year.

We have added the year in revised manuscript in line 266-267.

*"...the $CO_2$ influx rapidly returns to the same rate as in the control simulation at the 5th year after termination (Fig. 2b).*

L258 This seems like a trivial equation to provide, it's just a depth integral.

Agreed. The equation has been removed.

L308 I would avoid describing a global pH level as "healthy".

Thank you. We have changed the words as follows in line 338-340.

*"... Under the high-emission SSP585 scenario, surface pH declines rapidly from a relatively high level (pH = 8.03) to a more acidic state (pH = 7.67) by 2100."*

L332 The figure ordering is strange with respect to the text.

We double checked the ordering of all the figures and their apparence in the text, and have made sure that they are consistent.

L334-335 Does this mean the reductions in atmospheric air temperatures are not proportional to OAE? This is an important finding and requires discussion which appears to be absent. Why do the authors think this is the case? Is this because of internal variability? Are larger ensemble sizes of each experiment required?

Please reply to this comment in "General Comments" parts and have copied the content as follows:

*"...We also find that reductions in surface air temperature are not proportional to the level of alkalinity addition. This is because the slight cooling induced by OAE is smaller than the interannual variability simulated by the model, and is therefore obscured by internal climate variability (Lenton et al., 2018). We believe this*

*phenomenon warrants further investigation with larger ensembles or longer*

*simulations to confirm its robustness."*

L342 So the reductions in atmospheric CO₂ are consistent with the extent of OAE but not the reductions in surface temperatures? Please discuss, perhaps the temperature values are type errors, it's hard to see differences in figure 3.

Please see our reply to your previous comment. We have also redrew figures and rewording the the legend.

"…

[Figure]

*Figure 2: Temporal changes of (a) mean alkalinity in the upper 100 m (unit: meq/m³), (b) CO₂ influx*

*(unit: mmol/m²/yr), (c) atmospheric CO₂ (unit: ppmv), (d) integrated DIC inventory (unit: Pmol),*

*(e) surface pH, (f) surface air temperature (unit: °C). Perpendicular grey dash lines in the year of 2050 denote the onset of the 7.5×, 10× alkalinity enhancement scenarios (OWE 75 and OWE10, respectively), as well as the termination of alkalinity addition (OWE0). The coloured dash lines in (c), (d), (e), (f) are the anomaly between OAE simulations and the control run."*

L367-368 It's primarily due to the transport of water masses into the subsurface prior to full-equilibration.

Agreed. The characteristics of the water mass is also related to the location where the OAE is deployed. And different deployment methods of OAE also affect the dissolution rate of alkalinity, thereby influencing the efficiency of OAE. In this section, we have included information about water masses, thus making the discussion more comprehensive. In line 402-404:

*"...The wide range in previous studies is probably due to spatial and temporal variability, as well as differences in OAE application methods, which will influence the contact time between the water mass and the atmosphere and the time for water mass to reach equilibrium."*

L383-375. Can the authors explain the role of the simulation time? Is this because of sediment feedbacks? Most ESMs lack such feedbacks anyway (see Planchat et al., 2023) so I'm not sure running the models for longer would make a difference.

Köhler (2020) demonstrate that the calcite saturation horizon and lysocline transition zones in sediment will deepen under OAE, which finally lead to an increase of $CaCO_3$ accumulation. This process extracts alkalinity from the ocean and reduces the efficiency of OAE. However, as you mentioned, most of the Earth System models did not consider the sediment processes in alkalinity cycle. We have added some discussions in this part.

In line 412-415:

*"...Most ESMs do not take into account sediment processes, or they treat sediment processes as a part of the closed calcium carbonate cycle without considering the complex processes of sedimentation (Planchat et al., 2023). The absence of sedimentation processes may lead to an overestimation of the efficiency of OAE on a longer time scale."*

And in line 424-426:

*"...Although the short simulation in He and Tyka (2023) and our study likely missed the decline stage in adsorption efficiency in Köhler (2020), but the lack of sediment processes will overrate the efficiency later than 2100."*

L386-389 Are these differences in efficiency robust? Have similar effects been detailed in other studies and if so, can the authors explain the mechanism controlling this?

We believe these differences in efficiency are robust. In previous studies, the efficiency of OAE along the coastal regions would show a rapid increase in the initial years, and then the growth rate would slow down, reaching a relatively slow efficiency growth rate or a stable efficiency level (e.g. He & Tyka, 2023). We believe that the lower efficiency in OWE10 is due to the increased magnitude of OAE. It has not yet reached a relatively stable efficiency stage by the end of this century, and thus its efficiency is slightly lower compared to the other two groups of experiments. However, we did not run the simulation from later than 2100, thus we cannot give the final efficiency.

L398-400 Be clear that Zhou et al perform OAE locally in all grid cells and don't rely on rivers for delivery.

Thank you. We have clarified the applying method of OAE in Zhou et al. (2024) in line 439-451.

*"...Moreover, Zhou et al. (2024)reported that absorption efficiency is higher when OAE is applied in the equatorial Pacific than in subtropical regions. In contrast, our simulations show low absorption efficiency in the equatorial Pacific and only minimal increases in DIC inventory. We attribute this discrepancy to differences in the calculation methods. Zhou et al. (2024) applied OAE regionally, adding alkalinity to all grid cells within selected regions, and defined efficiency as the ratio of the global increase in DIC inventory to the total alkalinity added. Their finding of high efficiency in the equatorial Pacific is intuitive, as upwelling there spreads additional alkalinity across the surface ocean, enhancing $CO_2$ uptake. By contrast, in the subtropical gyres, which is characterized by convergence, added alkalinity is more readily subducted into the deep ocean, reducing efficiency. In our approach, however, efficiency is calculated locally as the ratio of the increase in DIC inventory to the increase in alkalinity within the same region when compared the regional efficiency. Under this definition, strong upwelling in the equatorial Pacific promotes $CO_2$ outgassing, resulting in lower efficiency."*

L458 How do these rates of acidification and carbon uptake compare to those in the CTL simulation?

We have calculated the pH decrease rate as the indicator of the acidification rate. We find the acidification has accelerated in OWE0 simulation after the termination of OAE with a rate of 0.0054, faster than 0.0047 (from 2020 to 2100) and 0.0053 (from 2050 to 2100) in control run. However, the carbon uptake rate (the influx of $CO_2$, see Fig. 2b) decrease to the similar rate with control run under OWE0 at the 5th year after OAE termination.

L487-489 Indicative that even riverine OAE results in loss of non-equilibrated water masses from the surface ocean, which are equilibrated of ocean circulation timescales of centuries.

Thank you for your suggestion. We have added these discussions in our revised manuscript in line 533-534:

[revised manuscript text omitted]

https://doi.org/10.1038/s41558-024-02179-9

**Response to Reviewer #2**

We sincerely thank the reviewer for the thorough evaluation of our manuscript and constructive suggestions provided. We have carefully considered each comment and made corresponding revisions to improve the clarity, accuracy, and overall quality of the work. Below, we provide a detailed, point-by-point response to all comments.

Reviewer comment: "Zhu et al. (2025) discussed river-based ocean alkalinity enhancement (OAE) for carbon dioxide removal in Earth system models. The main innovation of this paper lies in its specific focus on river-based OAE, distinguishing it from previous studies that typically assumed OAE on a broader scale, such as in open ocean basins (i.e., Lenton et al., 2018) or coastal areas (He and Tyka, 2023). Additionally, in contrast to global studies that cover estuarine regions (e.g., Zhou et al. 2024), this study uniquely utilizes an emission-driven Earth System Model (ESM), which provides an opportunity to further investigate atmospheric feedback effects (Tyka, 2025). However, the manuscript appears to capture such feedback but does not yet attempt to further distinguish and discuss these atmospheric feedback effects. Refining this section would enhance the scientific significance of the paper. Furthermore, there is still room for improvement in the figures and presentation. I will provide specific suggestions for improvement in the following sections. Overall, the conceptual foundation of this research is solid, and revisions and improvements would make this paper a valuable contribution to the growing body of literature on ocean alkalinity enhancement models."

We sincerely thank the reviewer for the thoughtful and encouraging comments regarding the novelty and conceptual foundation of our work. We appreciate the recognition of our study's specific focus on river-based OAE and its use of an emission-driven Earth System Model (ESM), and we agree that further clarification and discussion of atmospheric feedback effects would enhance the manuscript's scientific value.

Reviewer comment: "However, the manuscript appears to capture such feedback but does not yet attempt to further distinguish and discuss these atmospheric feedback effects. Refining this section would enhance the scientific significance of the paper. Furthermore, there is still room for improvement in the figures and presentation. I will provide specific suggestions for improvement in the following sections."

We appreciate this suggestion and have expanded the discussion on atmospheric feedback effects in the Results and Discussion sections. Specific points are addressed in our responses to your individual comments.

In addition, we have improved the figures and their presentation by adding anomalies, enlarging labels and legends, and implementing other enhancements in line with the reviewer's recommendations.

Reviewer comment: "Additionally, I would like to share an idea with the authors:
Given that both Zhu et al. (2025) and Zhou et al. (2024) used the CESM2 framework
but with different atmospheric components, and considering that Zhou et al. (2024)
provide an OAE efficiency budget for various global regions, converting Zhou et al.
(2024)'s open-source results to the same OAE injection areas as in Zhu et al. (2025)
would not require significant additional work. However, this approach could provide
potential insights into the differences in OAE budgets due to atmospheric forcing and
feedback effects. Please note that this is beyond the scope of this review, and the
authors are not required to address this suggestion in the revision."

We thank you for this insightful suggestion. Indeed, we used the same CESM2
framework as Zhou et al. (2024). However, Zhou et al. (2024) forced their model with
historical atmospheric $CO_2$ concentrations from the Japanese 55-year Reanalysis
dataset (JRA55) and assumed that the OAE perturbation in their simulations was too
small to generate significant changes in atmospheric $CO_2$. In contrast, our simulation
employed a fully coupled, emission-driven $CO_2$ forcing under the esm-ssp585
scenario. This key difference in model configuration means the two studies are not
directly comparable. Nevertheless, we will consider your suggestion and endeavor to
incorporate such comparative analyses in future work.

195: The subtropical gyres seem to contribute to two distinct ventilation regions in
around 30°N and 30°S, which are analyzed in the Discussion section (paragraph at
406) but are not mentioned in the Results. Furthermore, Fig. 3 shows that the
distribution of ALK across global ocean basins is inconsistent. For example, the ALK
excesses in the North Atlantic is significantly stronger than in the North Pacific.
Therefore, it would be helpful to calculate the contents in Fig. 4 separately for the
Atlantic, Pacific, and Indian Oceans. The same approach is also recommended for the
DIC analysis in Fig. 8.

We thank the reviewer for this helpful suggestion. We have added the corresponding
results in the section 3.1 and discussed more on the vertical anomaly of alkalinity
along 150°W (PAC), 30°W (ATL) and 90°E (IND) transects under OWE10
simulation as a case in this reply and revised manuscript. Figures are shown in this
reply and in a supplementary file. In line 211-218 in our revised manuscript:

"...

*The distribution of alkalinity across global ocean basins exhibits heterogeneity. In the*
*Pacific Ocean, a positive anomaly of alkalinity is observed within the upper 300-400*
*m and penetrates deeper in both north and south subtropical gyres (Fig. S2a). The*
*increase in alkalinity is greater in the north Pacific than in the south Pacific. The*
*alkalinity anomaly in the Atlantic dominates the vertical distribution of zonal mean*
*alkalinity anomaly, as there has the highest alkalinity increase and deepest*
*penetration especially in SPNA as well as in subtropical gyres (Fig. S2b). In the*

*Indian Ocean, the positive alkalinity anomaly also extends to greater depths within*
*the subtropical gyres (Fig. S2c)."*

[Figure]

Figure S2. Vertical distribution of alkalinity anomaly along specific transects. (a)
Differences between OWE10 and CTL along 150°W, represent the changes in the
Pacific Ocean; (b) Differences between OWE10 and CTL along 30°W, represent the
changes in the Atlantic Ocean; (c) Differences between OWE10 and CTL along 90°E,
represent the changes in the Indian Ocean.

Fig. 3: It is recommended to use the anomaly for panels c-f, especially panel f. The
differences between the curves in the current version are too small, which affects
readability.

Thank you. We have added the anomaly in panels c-f and updated the figure in the
revised manuscript.

"...

[Figure]

*Figure 2: Temporal changes of (a) mean alkalinity in the upper 100 m (unit: meq/m3), (b) CO2 influx (unit: mmol/m2/yr), (c) atmospheric CO2 (unit: ppmv), (d) integrated DIC inventory (unit: Pmol), (e) surface pH, (f) surface air temperature (unit: ºC). Perpendicular grey dash lines in the year of 2050 denote the onset of the 7.5×, 10× alkalinity enhancement scenarios (OWE 75 and OWE10, respectively), as well as the termination of alkalinity addition (OWE0). The coloured dash lines in (c), (d), (e), (f) are the anomaly between OAE simulations and the control run.''*

269-282: The DIC decrease in the Southern Ocean and equatorial Pacific is not mentioned? This is an apparent phenomenon, and overlooking it is weird. This may be due to the atmospheric feedback effects present in the ESM, for example, the ALK injection in the northern oceans reducing PCO2atm, which could have led to net outgassing in the Southern Ocean. However, further analysis is needed to confirm this. A more detailed analysis of the atmospheric and surface ocean PCO2 outputs for both the CTRL and experimental groups is necessary to determine whether the DIC decrease is due to atmospheric feedback or other mechanisms.

We thank the reviewer for highlighting this important point. Our analysis shows that in the Southern Ocean, the difference between atmospheric and surface ocean $pCO_2$ is smaller in the OAE simulations than in the control run (Fig. S4), indicating enhanced outgassing under OAE. This mechanism likely explains the observed DIC decrease and becomes more pronounced with higher levels of alkalinity addition (Fig. S3), consistent with the reviewer's expectation.

However, this explanation applies only where the atmospheric $CO_2$ decrease exceeds the corresponding seawater $pCO_2$ decrease. In the equatorial Pacific, the OAE-induced reduction in seawater $pCO_2$ is comparable to that in atmospheric $CO_2$, resulting in only a slight reduction in outgassing and a small net increase in DIC inventory. We have added these clarifications to Section 3.3 of the revised manuscript in line 296-302.

*"...In contrast, the Southern Pacific exhibits only a modest increase. A slight reduction of DIC inventory is observed in the Southern Ocean under the three continuous OAE simulation relative to the control, with the intensification of this reduction under higher alkalinity addition levels (Fig. 6b-d and Fig. S3). This phenomenon is attributable to the fact that OAE effectively lowers atmospheric $CO_2$ concentrations, thereby inducing an enhanced outgassing in the Southern Ocean and ultimately leading to a net DIC inventory loss there (Fig. S4)."*

[Figure]

Figure S3. Anomaly of DIC inventory. (a) difference between OWE75 and OWE5; (b) difference between OWE10 and OWE5.

[Figure]

Figure S4. Anomaly of partial pressure of $CO_2$ between OAE simulations and the CTL in the last 10 years of the end of simulation. The partial pressure difference between the ocean and atmosphere is calculated by subtracting the pressure of $CO_2$ in the ocean from the pressure of $CO_2$ in the atmosphere.

Fig8: See suggestion for 195.

Thank you. We have added further discussion on the vertical anomaly of DIC concentration along 150°W (PAC), 30°W (ATL) and 90°E (IND) transects in this reply and revised manuscript in line 312-318.

*"…from the surface to 200–300 m depth. The vertical anomaly of DIC concentration across global ocean basins generally mirror the pattern of alkalinity anomaly. The net $CO_2$ uptake induced by alkalinity injection results in DIC increase in most of ocean basins. The subtropical gyres in all three basins facilitate the downward transport of newly absorbed DIC, leading to the positive DIC anomalies in deeper layers (Fig. S5a-c). However, unlike alkalinity, a reduction in DIC concentration is evident in the high-latitude regions of the Southern Hemisphere, consistent with the DIC inventory changes."*

[Figure]

Figure S5. Vertical distribution of DIC concentration anomaly along 150°W (PAC), 30°W (ATL) and 90°E (IND) transects. (a) Differences between OWE10 and CTL along 150°W, representing the changes in the Pacific Ocean; (b) Differences between OWE10 and CTL along 30°W, representing the changes in the Atlantic Ocean; (c) Differences between OWE10 and CTL along 90°E, representing the changes in the Indian Ocean.

320: It is recommended to provide a more detailed explanation for the increase in the North Atlantic in OWE0, particularly in Hudson Bay and the Northwestern Channel. Additionally, a noticeable pH increase is also observed in the Ross Sea. Is this related to sea ice or outgassing? Further analysis and clarification would be beneficial.

We thank the reviewer for this valuable suggestion. In Hudson Bay and the Northwestern Channel, alkalinity accumulates during the first 30 years of OAE (Fig. S1). The narrow passages in these regions restrict exchange with the open ocean, allowing alkalinity to persist locally. After OAE termination, this accumulated alkalinity acts as a residual "source" that is advected to the SPNA, sustaining elevated alkalinity and pH in both Hudson Bay and the SPNA relative to the control run.

In the Ross Sea, the observed pH increase also reflects higher alkalinity. The model shows a modest alkalinity rise in the 2090s (Fig. S1), likely due to upwelling of water masses carrying excess alkalinity originating from earlier OAE. This upwelled alkalinity is the most plausible driver of the elevated pH in this region.

We have incorporated these explanations into Sections 3.1 and 3.4 of the revised manuscript in line 198-203, and will include the supporting figures in the supplementary material:

*"...The accumulation and later release of alkalinity in the Hudson Bay is a potential reason why alkalinity in the SPNA remains relatively high even after alkalinity enhancement has ceased. During the first 30 years of OAE, alkalinity is accumulated and retained in this region due to the narrow passages in Northwestern Channel and Hudson Bay. When the OAE terminated, this accumulated alkalinity becomes a new "source", which is transported to the SPNA and effectively maintains the alkalinity compared to the control group (Fig. S1)."*

And in section 3.4, we have explained the increase of pH in SPNA and Ross Sea. In line 352-357:

*"...As expected, greater alkalinity additions correspond to stronger pH buffering.*

*Although alkalinity input ceases in OWE0, this scenario still shows a slight increase in surface pH compared to the control, particularly in the SPNA (Fig. 8e) where there are pronounced alkalinity increase compared to the control. However, this increase is considerably smaller than those observed in the continuous OAE treatments. We also find an increase of pH in the Ross Sea by the end of this century, which is attributable to the upwelling-mediated return of OAE-induced alkalinity to the surface, thereby elevating surface pH (Fig. S6)."*

[Figure]

Figure S1. Alkalinity anomaly in OWE0 compared to CTL during simulation phase.

[Figure]

Figure S6. Surface pH anomaly in OWE0 compared to CTL during simulation phase.

332-334: The result is reasonable, but comparing the temperature values at a single time point is not appropriate. It is recommended to use the average temperature over the last 10 years or a similar metric for the analysis.

We appreciate the reviewer's comment. We actually compared the average temperature of the last 10 years between CTL and other OAE simulation. We have revised our manuscript in line 367-369 and make this comparison much clearer.

*"All the four OAE treatments show a slight decrease of temperature in the last 10 years of this century, with 0.45 °C in OWE5, 0.39 °C in OWE75, 0.34 °C in OWE10, and 0.31 °C in OWE0 compared to CTL (Fig. 2f)."*

417-438: Recommend to streamline this section, as it currently appears more like a literature review rather than a targeted discussion.

We thank the reviewer for this constructive suggestion. In the revised manuscript, we have streamlined the paragraph to focus on the key challenges relevant to our study, while retaining only the most essential contextual references. The revised text now emphasizes the implications of alkalinity loss and material constraints for river-based OAE, rather than providing an extended literature survey. This change makes the discussion more concise and targeted. In line 463-472:

*"One of the most critical challenges in OAE is alkalinity loss through precipitation, which can rapidly reduce efficiency (Moras et al., 2022). The extent of this loss depends on the type and form of added material, solution state, and presence of particles (Hartmann et al., 2023). For riverine OAE, substantial losses may occur in estuaries, making it essential to regulate addition rates. $CO_2$-equilibrated alkaline solutions and certain Mg-rich minerals can help limit precipitation(Jones, 2017; Pan et al., 2021), though some, like olivine, may still be less efficient due to particle-induced losses (Fuhr et al., 2022). Using finely ground particles can improve dissolution but increases energy costs, while particles in river plumes can promote heterogeneous precipitation (Wurgaft et al., 2021). These factors highlight the need for careful material selection and delivery design to minimize losses in real-world applications."*

Minor comments:

Figures 3-8:The numbering of the subplots, the legend, and the labels have fonts that are too small and need to be enlarged. Figures suffer from low image resolution, which affects readability. Please ensure that the images in the final published version are clear.

We have enlarged the labels and legends in our revised manuscript. And we will make sure the images clear in our final version.

203:50-70N? There is a difference with Fig. 5d.

Agreed. We have revised the latitude range in line 219-220.

*"...Although alkalinity addition ceases after 2050 in the OWE0 simulation, a positive alkalinity anomaly persists through 2100, reaching depths of 1500 m near 50°– 70°N."*

226: It is suggested to indicate the time of comparison here or in the caption of Fig. 6. Although it is provided later, it has not been explained earlier.

Agreed. We have revised our manuscript accordingly in 246-249:

*"...OAE modifies the air-sea $CO_2$ gradient, promoting greater $CO_2$ absorption in areas where the ocean is undersaturated and diminishing $CO_2$ release in regions where it is supersaturated (Fig. 5). This results in a net increase in ocean carbon storage and contributes to a reduction in atmospheric $CO_2$ levels."*

245: Considering that the rate of rebound is rapid, "rapidly returns" would be more appropriate than "eventually returns".

We appreciate for your suggestion. We have changed "eventually returns" to "rapidly returns" in revised manuscript in line 265-267.

*"...When alkalinity addition ceases in 2050 (OWE0), the $CO_2$ influx rapidly returns to the same rate as in the control simulation (Fig. 2b).*

392: Fig 4, not 3.

Thank you. We have corrected this. We have deleted the Fig.1 in our revised manuscript according to the Reviewer #1, thus it finally is Fig. 3. In line 432-433

*"...Although alkalinity is introduced via rivers, its effects extend to the open oceans, with more pronounced impacts observed in the Atlantic and Indian Oceans compared to the Pacific (Fig. 3)."*

395: The logic seems unclear. It is recommended to rephrase as: Compared to the North Atlantic, the western boundary current of the North Pacific occurs outside the island chains, and a large amount of ALK excess is enriched inside the island chains, preventing it from spreading to the wider Pacific.

Thank you for the helpful recommendation. We have clarified our expression in line 434-439.

*"...For instance, in the Atlantic, excess alkalinity from the Caribbean Sea can be transported to the North Atlantic by the Gulf Stream, a strong western boundary*

*current. Compared to the North Atlantic, the western boundary current of the North*
*Pacific occurs outside the island chains, and a large amount of ALK excess is*
*enriched inside the island chains, preventing it from spreading to the wider Pacific."*